# Photosynthetic Nutrient and Water Use Efficiency of *Cucumis sativus* under Contrasting Soil Nutrient and Lignosulfonate Levels

**DOI:** 10.3390/plants10020340

**Published:** 2021-02-10

**Authors:** Elena Ikkonen, Svetlana Chazhengina, Marija Jurkevich

**Affiliations:** 1Karelian Research Center RAS, Institute of Biology, Puskinskaja, 11, 185610 Petrozavodsk, Russia; svirinka@mail.ru; 2Karelian Research Center RAS, Institute of Geology, Puskinskaja, 11, 185610 Petrozavodsk, Russia; chazhengina@mail.ru

**Keywords:** cucumber, pulp and paper industry wastes, biomass, photosynthesis, nutrient concentration

## Abstract

To reduce the use of commercial conventional inorganic fertilizers, the possibility of using pulp and paper industry wastes in agriculture as an alternative source of nutrients is recently under study and discussion. This work aimed to evaluate the effect of sodium lignosulfonate application to soil on photosynthetic leaf nutrient (N, P, K, Ca, Mg, Fe, Mn, and Na) and water use efficiency. A pot culture experiment was conducted with cucumber seedlings, using five lignosulfonate concentrations (0, 1, 2.5, 5, and 10 vol. %) in sandy soil under sufficient or low nutrient availability for plants. The impact of nutrient availability on the plants’ physiological traits was stronger than the lignosulfonate impact. Under sufficient nutrient availability, the lignosulfonate application resulted in decreased photosynthetic N, P, K, Ca, Mg, Fe, and Na use efficiency. Cucumber growth and development, and photosynthetic nutrient, water, and light use efficiency were significantly reduced with a nutrient deficiency. The sodium lignosulfonate application was not successful in eliminating the negative effects of nutrient deficit on cucumber seedlings.

## 1. Introduction

Plant growth and development largely depend on soil nutrient availability. A wide range of soils are noted for their low nutrient-supplying capacity, and for agricultural purposes they need to be improved with commercial high-cost fertilizers. New economic instruments of improving soil fertility and reducing the cost of agricultural products are currently under study and discussion [1,2,3]. The possibility of using of pulp and paper industry wastes as an alternative source of soil nutrients is also discussed [4,5].

Lignosulfonates (LSs) are by-products of pulp and paper mills producing cellulose and are considered to be a special class of industrial lignin. Although LSs are increasingly used in industry, including in animal feed, pesticides, and soil improvers [4,5,6], they continue to remain underutilized [7,8]. A wide variety of elements are presented in carboxylic, phenolic, sulfonic, and other groups of LS’s structure [9]. Recent studies have shown that LSs can be applied as a soil conditioner and chelate fertilizer [10,11], improving the physical, chemical, and biological characteristics of soils [12,13]. For plants, it was reported that LSs can enhance callus proliferation rate and adventitious root formation [14] and improve root activity [15]. The LS-related stimulation of plant growth was shown for some species [16,17,18], but the opposite effect has been also found. Stapanian and Shea [19] reported that the biomass of woody vegetation was not affected by LS, and biomass of herbaceous plants was significantly decreased with increasing content of LS in the soils. Ertani et al. [16] suggested that LSs play a positive role in the photosynthetic process because its application can increase chlorophyll content and rubisco enzyme activity, as have been found for *Zea mays*. However, no information is available concerning the effects of LS on plant photosynthetic activity and photosynthetic nutrient use efficiency, defined as the net photosynthetic rate per unit leaf nutrient content.

The ability of LSs to improve soil chemical parameters [10,13] and chelate nutrient ions [20] allows suggesting that LS can enhance leaf nutrient content, especially when plants grow in soil characterized by low natural fertility. However, there has been no published evidence supporting this claim. It can be hypothesized that LS-related modification of leaf nutrient content could affect photosynthetic capacity and/or efficiency of nutrient use. Leaf N and P are both essential nutrients involved in the photosynthetic process, but other leaf nutrients, such as K, Ca, Mg, Fe, Mn, and Na, can modify the photosynthetic capacity [21,22] and affect the photosynthetic efficiency of using of other nutrients, mainly, N and P, as has been shown recently [2,23].

Our study aimed to investigate the LS effects on physiological traits including plant growth, photosynthesis, leaf content of N, P, K, Ca, Mg, Fe, Mn, Na, and photosynthetic nutrient, water, and light use efficiency in *Cucumis sativus* L. seedlings. Comparisons were made using seedlings grown under two levels of soil nutrient availability (sufficient and low). We assess whether the LS amendment would improve the physiological state of plants, and under which condition of soil nutrient availability these improvements could be more pronounced.

## 2. Results

### 2.1. pH and Nutrient Content of Sodium LS and Soil Substrates

The pH value and the content of N, P, K, Ca, and Mg in the sodium LS used in this study are shown in the Table 1. Following the increase in soil LS content, N, K, Ca, Mg, and Na concentrations increased, and soil pH values, P, Fe, Mn and Al remained unchanged (Table 2). While the 10LS substrate had 44 and 64% higher content of N and K, respectively, than the 0LS one, LS resulted in increased Ca, Mg, and Na concentrations by 3, 5, and 9 times, respectively.

### 2.2. Plant Biomass

The dry biomass and leaf number and area were significantly lower in seedlings grown under the condition of low nutrient availability (LNA) than sufficient nutrient availability (SNA) (Table 3). By contrast, the leaf mass per unit leaf area (LMA) were significantly higher in LNA-grown seedlings compared to their SNA-grown counterparts.

The LS application caused a significant decrease in total plant and leaf biomass only in 10LS seedlings grown under SNA. For LNA-grown seedlings, LS had no significant effect on biomass accumulation and its allocation into organs as well as leaf number and area. The LMA values increased following the increase in substrate LS content regardless of nutrient conditions.

### 2.3. Gas Exchange and Chlorophyll Parameters

As expected, the decline in nutrient availability for the cucumber seedlings dramatically decreased both the *F*_v_/*F*_m_ and SPAD index, as well as the *A*_n area_ and *A*_n mass_ rates, *g*_s_, and Tr values (Table 3). The LNA-grown seedlings had higher values of *C*_i_:*C*_a_ ratios than the SNA-grown ones. The LS application did not affect the *F*_v_/*F*_m_ values regardless of nutrient conditions, but significantly decreased the SPAD index for 1LS and 2.5LS, the *A*_n area_ rate for 1LS, *A*_n mass_ for 2.5LS and 5LS seedlings grown under SNA. For the LNA seedlings, both *A*_n area_ and *A*_n mass_ rates tended to increase in the cases when LS was added to the soil, but these increases were not large enough to be statistically significant. The LS did not alter the *C*_i_:*C*_a_ ratios in the SNA leaves and decreased these parameters in LNA ones. As shown in Table 4, the *g*_s_ and Tr values were significantly affected by nutrient availability, but not LS application.

### 2.4. Leaf Nutrient Content

The decline in nutrient availability caused an increase in N, Fe, Na, and Al content in 0LS leaves by 20%, 21, 3.6, and 5.8 times, respectively, and a decrease in P, K, and Mg content by 38, 28, and 34%, respectively (Table 3). According to the TWO-WAY ANOVA, the effect of LS application was significant for leaf K, Ca, Mg, Fe, Mn, Na and Al (Table 4). The N and P values were the highest in 1LS leaves among all LS treatments under the SNA condition. Following the increase in soil LS, leaf Na content increased, Mn content decreased, and K content did no change regardless of nutrient availability. Under both nutrient conditions, the 1LS seedlings had higher and 10LS seedlings had lower leaf concentrations of Ca and Mg than their 0LS counterparts. While the increase in the substrate LS content decreased leaf Fe accumulation under LNA, this tendency was not recorded for seedlings grown under SNA. Among SNA-grown seedlings, the content of Al was the lowest in 2.5LS and 5LS leaves. For LNA-grown seedlings, by contrast, LS increased leaf Al content.

The decline in nutrient availability caused an increase in the concentration of N, Ca, Mn, Fe, and Na in the unit leaf area (SLN, SLCa, SLMn, SLFe, and SLNa accordingly in Table 3), as has been found for 0LS seedlings. The significant effect of LS was found for most of the studied nutrients when their concentration was calculated per unit leaf area (Table 4).

### 2.5. Photosynthetic Nutrient Use Efficiency

For all elements under study, photosynthetic nutrient use efficiency was lower in LNA-grown seedlings than their SNA counterparts (Figure 1). The LS effect on the efficiency of using of nutrients depended on the soil nutrient conditions. While for the photosynthetic N use efficiency (PNUE), photosynthetic P use efficiency (PPUE), photosynthetic K use efficiency (PKUE), and photosynthetic Fe use efficiency (PFeUE) values, the TWO-WAY ANOVA revealed an insignificant effect of the LS treatment (*p* > 0.05).

PCaUE, PMgUE, PMnUE, and PNaUE were significantly altered by both LS application and nutrient availability (*p* < 0.01, Table 4). Among most of the studied elements, no significant effect of LS on photosynthetic nutrient use efficiency was observed for LNA-grown seedlings. For SNA-grown seedlings, LS decreased PNUE, PPUE, and PNaUE on average by 74, 114, and 79%, respectively. The PKUE, PCaUE, PMgUE, and PFeUE values were significantly lower in 1LS and 2.5LS seedlings compared to their 0LS counterparts. Unlike other nutrients under the study, LS enhanced PMnUE values regardless of conditions of nutrient availability (Figure 1g).

### 2.6. Photosynthetic Water Use Efficiency and Apparent Quantum Yield

The decline in nutrient availability caused a decrease in PWUE and α value in cucumber leaves (Figure 1i,j). Both PWUE and α were significantly altered by LS application as well as nutrient availability (Table 4). The LS effect on PWUE was stronger under the LNA than SNA condition. Thus, for the LNA-grown seedlings, the PWUE values were, on average, 2.3 times higher in the seedlings grown on the substrates containing LS than substrates without LS. Among the SNA-grown seedlings, the lowest α value was found in 1LS leaves. Under the LNA condition, the LS application resulted in increased α by 4.5 times.

## 3. Discussion

In this study, we sought to understand how the application of sodium LS affects physiological state, particularly photosynthetic nutrient, water, and light use efficiency in *C. sativus* seedlings. In order to understand whether the response of these parameters to the LS application depends on nutrient conditions, the cucumber seedlings grown under sufficient or low nutrient availability (SNA or LNA, respectively). We found that the impact of nutrient availability on plant physiological traits was much stronger than the LS impact. The decline in nutrient availability for seedlings resulted in the suppression of their growth and development, photosynthetic activity, and stomata opening (Table 3), as well as decreased photosynthetic nutrient use efficiency (Figure 1). As outlined in the Introduction we hypothesized that sodium LS application can improve soil nutrient availability and nutrient uptake by plants that, therefore, can positively affect the plant physiological traits. Our hypothesis was based on the early findings that LS can improve soil chemical conditions [9,10] and some physiological traits of plants [14,15,16,17,18]. The agricultural application of LSs, as low-cost compounds, could be highly advantageous [6,10,11] and might promote the effective utilization of wastes of paper and pulp industry. However, we found no evidence of beneficial effects of sodium LS on cucumber seedlings’ growth, development, photosynthesis, and photosynthetic nutrient and water use efficiency, regardless of the conditions of nutrient availability for the plants. Under the sufficient nutrient condition, LS did not promote plant nutrient uptake, and even resulted in some toxic effect, as well as decreased efficiency of nutrient use at leaf level. Moreover, LS application was not successful in eliminating the negative effects of nutrient deficiency, accompanied by high soil acidity, on plant physiological traits, including photosynthetic nutrient and water use efficiency.

The results of this study are consistent with the well-documented negative effect of soil nutrient deficiency on plant growth and photosynthesis [22,23,24]. The decrease in nutrient and water use efficiency observed in this study is likely to be related to the strong depression of the photosynthetic process caused by nutrient deficiency. There is strong evidence that in addition to nutrient limitation, high soil acidity is also largely responsible for lower nutrient use efficiency [25]. Dramatically high levels of Fe and Al in cucumber leaves grown under LNA are likely to be caused by the low pH of the soil under study, which might increase the localized availability of these elements for plants. This effect was not found for the SNA-grown seedlings, mainly due to the increase in soil pH by nutrient solution. At lower levels of supply, Al stimulates root growth, increases nutrient uptake and plant resistance to stress, but in acidic soils, aluminum toxicity leads to the inhibition of plant growth, disturbs the transport of ions, and causes an imbalance of nutrients [26]. Moreover, soil Al saturation significantly affects leaf P and N resorption [27]. In addition to aluminum, iron and manganese toxicity also reduces root development and plant growth [28,29]. The high Mn concentration found in our study is considered to be toxic for many species, including cucumber [29]. Thus, the high concentration of Al, Fe and Mn in leaves of cucumber seedling, along with nutrient deficiency, can be one of main reasons for depression of photosynthesis and low nutrient and water use efficiency observed in this study. Since LS application caused a decrease in leaf Mn, as well as, Fe content, probably, due to LS-related decrease in soil acidity, LS could be assumed as a Mn and Fe detoxifier, but further research needs to examine this assumption.

Delgado et al. [30] reported that photosynthetic nutrient use efficiency (mainly PNUE and PPUE), is, in general, higher in soil with higher nutrient availability. Our results are consistent with this finding. One of the reasons of low nutrient use efficiency in plants grown under nutrient deficiency probably occurs due to higher LMA in these plants, compared to the ones grown under richer conditions, as has been found earlier for some species [31]. As well as nutrient deficiency, the LS application increased LMA regardless of the nutrient availability for the seedlings. The higher LMA can be connected with a higher fraction of leaf dry mass as a cell wall and a higher allocation of proteins in the cell wall. The cell wall proteins, which are an important component of non-photosynthetic N, can contribute to the negative correlation between LMA and PNUE [32]. Since the decline in nutrient availability caused a significant decrease in the *A*_n_ rate, but not leaf N content, it can be assumed that the majority of leaf N of the LNA-grown seedlings was not connected with the photosynthetic processes. Interestingly, for the seedlings grown under nutrient deficiency, the increase in LMA was related with the increase in PNUE. Moreover, in contrast to the study of Pons and Westbeek [33] who have reported that intercellular CO_2_ concentration contributed to the higher PNUE for some woody species, the results of our study showed a negative correlation between these parameters for LNA-grown seedlings. These results suggest that LMA and intercellular CO_2_ concentration are not always the major factors responsible for the variation in PNUE.

It is well documented that the mineral–nutrient status of plants plays a critical role in water use efficiency [34]. Our results are in agreement with the findings that sufficient nutrient availability is not only required for the photosynthetic process and plant growth but also can improve PWUE [35], as well as light use efficiency defined as the α value [36]. In contrast to studies suggesting a trade-off relationship between PNUE and PWUE [35], our results showed that the photosynthetic efficiency of the use of most nutrients decreased as well as PWUE in response to the decline in nutrient availability for plants.

Contrary to our expectations, we found that LS application did not enhance nutrient use efficiency in the seedlings grown under LNA. Besides, LS did decrease these parameters under SNA. This probably occurred because of an increase in Na content when the high concentration of LS was added to the soil. The LS-related increase in soil Na content is likely to cause soil sodium salinization. Appearing at high concentrations, Na ions can inhibit the main physiological processes in plants due to osmotic stress or ionic effects [37]. The increase in cellular Na can affect the acquisition and distribution of essential nutrients such as N, K, and Ca [38], and, also Mg, as our results have shown (Table 3). For the seedlings grown under sufficient nutrient conditions, the LS-related decrease in the photosynthetic efficiency of the use of most studied nutrients is likely to be associated with the strong increase in leaf Na content. Recent studies have also shown that the photosynthetic efficiency of the use of some nutrients can be modified by other leaf mineral nutrients. For example, Hou et al. [39] found that the variation of leaf K content can alter the PNUE values of rice leaves, and leaf Mg can enhance both PNUE and PPUE, as was shown for Karst plants [40].

Some positive impacts of LS application on cucumber seedlings can be seen through the increase in PWUE and α values for plants grown under a nutrient deficiency (Figure 1i,j). However, this impact was not strong enough to restore the level of water and light use efficiency of these seedlings close to the level of their SNA-grown counterparts. Such plant nutrients as K, P, Mg, Mn, and others can affect plant traits related to water status and improve PWUE of plants [41]. Although LS increased the soil concentration of K and Mg, we did record a decrease in the content of these nutrients in the leaves of LNA-grown seedlings. Thus, in our case, any other mechanism could be responsible for the LS-related increase in PWUE in LNA-grown seedlings.

## 4. Conclusions

The present study has highlighted that the impact of soil nutrient deficit on plant physiological traits can be much stronger than sodium LS impact. The decline of nutrient availability for plants resulted in increased leaf mass per area and decreased plant growth, nutrient uptake, and photosynthetic nutrient, water, and light use efficiency. The LS application to sandy soil was not successful in eliminating the negative effects of the nutrient deficit on the physiological state of the cucumbers. The results demonstrated that sodium LS decreases soil acidity and increases soil content of some nutrients: mainly, N, Ca, Mg, and Na. The LS-related increase in soil Na resulted in a strong increase in leaf Na to a toxic level, which could be one of the main reasons for the decreased photosynthetic nutrient use efficiency for the plants grown under sufficient nutrient availability.

## 5. Materials and Methods

### 5.1. Soil Substrate Preparation

For this pot experiment, we used sandy loam soil collected from the Korza valley, Karelia, the northwest of Russia. In its native state, this soil has only a poor-to-fair water holding capacity. Sandy soils contain very few nutrients and are especially deficient in nitrogen [42]. The humus content varied from 0.1 to 0.5%, total nitrogen was 0.01–0.03%, and soil pH was in the range of 3.5–6 [39].

The collected soil was air-dried at 21–23 °C for 14 days and sieved with a 2 mm sieve. The entire volume of the dry soil was divided into five parts and mixed with sodium lignosulfonate (Kondopoga Pulp and Paper Mill, Karelia, Russia) to achieve its concentration in the soil substrates, equal 0, 1.0, 2.5, 5.0, and 10 % (*v*/*v*), designated as 0LS, 1LS, 2.5LS, 5LS, and 10LS treatments, respectively. Lignosulfonate used in this study contained 48% of O, 42% of C, 7% of S, 4% of H with ash content of 17% [9]. All soil substrates were incubated under 21–23 °C and 70–80% of the maximum soil water holding capacity for 90 days. After the incubation period, the substrates were parked into plastic 0.80 L pots. Each treatment included sixteen pots.

### 5.2. Plant Material and Growth Conditions

Uniform seeds of cucumber (*C. sativus* L., var. Kurag) were imbibed in distilled water for 24 h and then sown with four seeds per pot. The pots were subjected to two controlled climate chambers (BKIII−73, Russia) with conditions maintained at 23/20 °C day/night temperature, 70% relative air humidity, 16-h photoperiod, 300 μmol m^−2^ s^−1^ of photosynthetic photon flux density (PPFD), and 410 ± 10 ppm of air CO_2_ concentration. The pots were randomly repositioned daily within each chamber, and weekly between the chambers to avoid plant location effects.

The pots of each LS treatment were divided into two blocks. For the first block of the pots, the nutrient solution was composed of 1 g L^−1^ Ca(NO_3_)_2_, 0.25 g L^−1^ KH_2_PO_4_, 0.25 g L^−1^ MgSO_4_ 7H_2_O, 0.25 g L^−1^ KNO_3_, a trace amount of FeSO_4_ with pH 6.2−6.4, and EC 2.0 mS cm^−1^ was supplied every two days. For the second block of the pots, water was supplied also every two days. The first block of plants was designated as plants grown under the condition of sufficient nutrient availability (SNA), and the second one was grown under low nutrient availability (LNA). One week after the sowing, the seedlings were thinned to one seedling per pot. The gas exchange measurements started when the seedlings were 24 days of old. The plant growth parameters were determined after the gas exchange and chlorophyll fluorescent measurements.

### 5.3. Plant Growth Parameters

The plants were harvested and the dry weight (at 70 °C to weight constancy) was determined of the plants separated into leaves, stems, and roots. Total leaf number per plant was counted, and the total leaf area per plant was determined by leaf scanning and using the program “AreaS”. The LMA values were calculated as leaf mass per unit leaf area. The leaf to total biomass ratio (g leaves per g plant) was calculated.

### 5.4. Gas Exchange Measurements

Net CO_2_ assimilation (*A*_n area_) rate on area basis, leaf transpiration rate (Tr), stomatal conductance (*g*_s_), ambient (*C*_a_) and internal (*C*_i_) CO_2_ concentration, and photosynthetic water use efficiency (PWUE = *A*_n_ Tr^−1^) were measured with a portable photosynthesis system (HCM-1000, Walz, Effeltrich, Germany) using the youngest fully expanded leaves. During gas exchange measurements, leaf temperature was kept at 25 °C, PPFD was 1200 μmol m^−2^ s^−1^, and relative air humidity varied from 60% to 65%. The irradiance response of net CO_2_ exchange was determined, starting at 1200 μmol m^−2^ s^−1^ of PPFD, followed by measurements at 1000, 800, 300, 60, 40, 20, and zero PPFD. The apparent quantum yield of photosynthesis (α) was calculated as the slope of *A*_n_ versus irradiance of 20, 40, and 60 μmol m^−2^ s^−1^. The chlorophyll fluorescence was measured using a portable chlorophyll fluorometer MINI-PAM (Walz, Effeltrich, Germany) and chlorophyll content was measured with a SPAD 502 (Minolta Corp., Ramsey, NJ, USA) as SPAD units on the same leaves used for the gas measurements. The maximal quantum yield of PSII photochemistry was calculated as *F*_v_/*F*_m_ = (*F*_m_ - *F*_0_)/*F*_m_; where *F*_m_ and *F*_o_ were maximum and minimum fluorescence of dark-adapted leaves, respectively. The values of *A*_n_ rates on a mass basis (nmol g_DM_ s^−1^) were calculated using measured values of leaf mass per area (LMA).

### 5.5. Chemical Analyses

After the incubation, the soil substrates were air-dried and sieved with a 1 mm sieve. For the chemical analyses three sub-samples of each LS substrate were used. The pH of soil substrates and LS was measured in 1 M L^−1^ KCl solution (pHKCl) at a ratio of 1:2.5 (w:w). Total N concentration in the soil substrates was determined by the Kjeldahl method using a Kjeltec analyzer [43], available K and P were extracted with 0.2M HCl and determined using spectrophotometric and flame photometric methods (SF 2000 OKB Spectrum, Saint-Petersburg, Russia) accordingly, following Kirsanov’s procedure used in Russia [44]. Soil Ca, Mg, Fe, Mn, Na, and Al were extracted with 1MNH_4_Ac buffered to pH 7 and then determined using spectrophotometric atomic absorption (Shimadzu AA-7000, Kyoto, Japan), following the procedure recommended by van Reeuwijk [45].

For the chemical leaf analyses, we used four seedlings of each treatment grown under conditions of sufficient nutrient availability, but for the seedlings grown under low nutrient availability, all leaves of the eight seedlings of each treatment were combined into one sample. Homogenized leaf samples of 0.2–0.3 g were digested with HNO_3_ and HCl (Vekton, Russia). Kjeldahl digestion was used for the determination of leaf N and P. The N content was determined by the nesslerization of the ammonia, and the P content was determined by the ammonium molybdate method. Both leaf N and P were analyzed by spectrophotometer (SF 2000 OKB Spectrum, Saint-Petersburg, Russia). Leaf element concentrations of K, Ca, Mg, Fe, Mn, Na, and Al were determined by spectrophotometric atomic absorption (Shimadzu AA-7000, Kyoto, Japan) in the Core Facility “Analytical laboratory” of the Forest institute of KRC of RAS. The chemical analyses of LS were carried out similarly to the chemical analyses of soil substrates.

The specific leaf element contents per unit leaf area were calculated using the values of LMA and the leaf element contents. Photosynthetic nutrient use efficiency was calculated by dividing area-related *A*_n_ (*A*_n area_) by the value of specific leaf element content.

### 5.6. Statistical Analyses

For each treatment, the means ± SE were determined with at least five and more replicates for the *A*_n_, *g*_s_, Tr, *C*_i_:*C*_a_, LMA, WUE, and α parameters. To assess the significant difference between the treatments at the *p* < 0.05 level, the least significant difference (LSD) of ANOVA was used with Statistica software (v. 8.0.550.0, StatSoft, Inc., Tulsa, OK, USA). The effects of soil LS concentration, nutrient availability and their interaction were analyzed using a TWO-WAY ANOVA.

## Figures and Tables

**Figure 1 plants-10-00340-f001:**
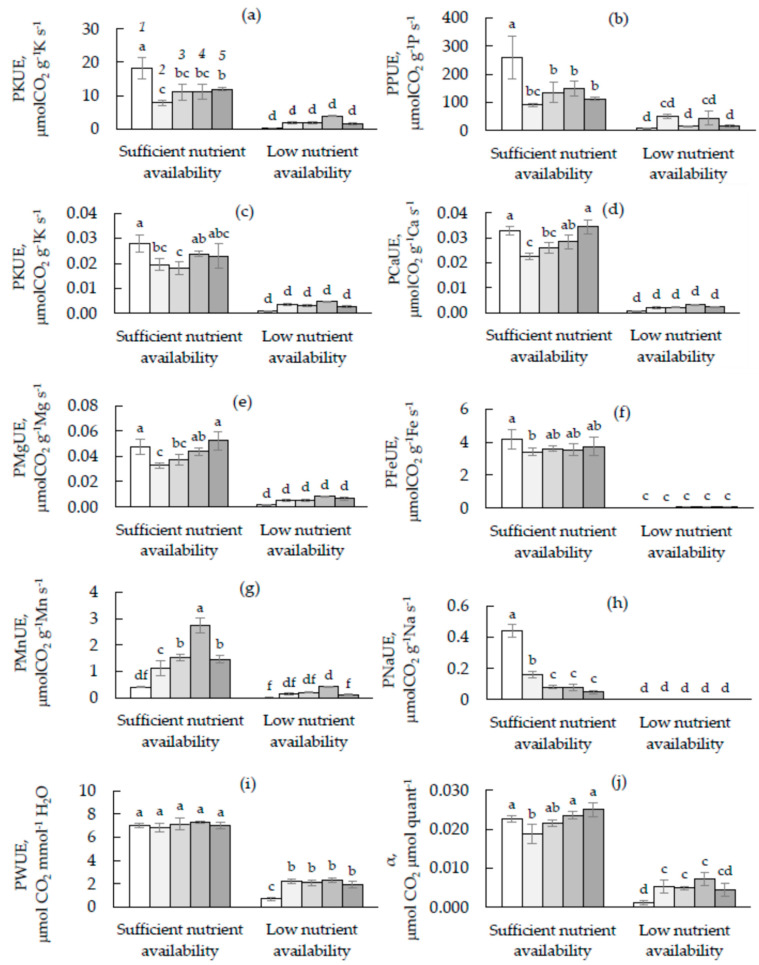
Photosynthetic nitrogen (PNUE, **a**), phosphorus (PPUE, **b**), potassium (PKUE, **c**), calcium (PCaUE, **d**), magnesium (PMgUE, **e**), iron (PFeUE, **f**), manganese (PMnUE, **g**), sodium (PNaUE, **h**), water (PWUE, **i**), and light (α, **j**) use efficiency for cucumber seedlings grown in the soil with a lignosulfonate concentration of 0 (*1*), 1 (*2*), 2.5 (*3*), 5 (*4*), and 10 (*5*) % under conditions of sufficient or low nutrient ability. Different letters indicate significant differences at *p* < 0.05.

**Table 1 plants-10-00340-t001:** The pH value and nutrient content in the sodium lignosulfonate used in the study.

pH	N, g kg^−1^	P, g kg^−1^	K, g kg^−1^	Ca, g kg^−1^	Mg, g kg^−1^	Na, g kg^−1^
4.61	1.59	0.13	0.12	0.75	0.76	2.19

**Table 2 plants-10-00340-t002:** The pH values and nutrient content in the soil substrates containing 0 (0LS), 1 (1LS), 2.5 (2.5LS), 5 (5LS), and 10 (10LS) % lignosulfonate.

Variables	0LS	1LS	2.5LS	5LS	10LS
pH	3.89 ± 0.23 ^a^	4.21 ± 0.29 ^a^	4.23 ± 0.30 ^a^	4.23 ± 0.21 ^a^	4.24 ± 0.18 ^a^
total N, g kg^−1^	1.05 ± 0.07 ^b^	1.43 ± 0.12 ^a^	1.45 ± 0.12 ^a^	1.48 ± 0.09 ^a^	1. 52 ± 0.11 ^a^
available P, g kg^−1^	0.11 ± 0.01 ^a^	0.12 ± 0.01 ^a^	0.13 ± 0.01 ^a^	0.11 ± 0.01 ^a^	0.11 ± 0.01 ^a^
available K, g kg^−1^	0.08 ± 0.01 ^bc^	0.10 ± 0.01 ^b^	0.12 ± 0.01 ^ab^	0.13 ± 0.01 ^a^	0.13 ± 0.01 ^a^
exchangeable Ca, g kg^−1^	0.07 ± 0.01 ^c^	0.16 ± 0.01 ^b^	0.20 ± 0.02 ^a^	0.20 ± 0.02 ^a^	0.24 ± 0.02 ^a^
exchangeable Mg, g kg^−1^	0.003 ± 0.000 ^c^	0.008 ± 0.001 ^b^	0.013 ± 0.001 ^a^	0.014 ± 0.002 ^a^	0.015 ± 0.002 ^a^
exchangeable Fe, g kg^−1^	19.9 ± 2.4 ^a^	19.3 ± 1.8 ^a^	19.1 ± 2.0 ^a^	18.3 ± 1.3 ^a^	21.3 ± 2.0 ^a^
exchangeable Mn, g kg^−1^	0.42 ± 0.05 ^a^	0.45 ± 0.04 ^a^	0.38 ± 0.04 ^a^	0.39 ± 0.05 ^a^	0.48 ± 0.04 ^a^
exchangeable Na, g kg^−1^	0.02 ± 0.00 ^c^	0.05 ± 0.01 ^b^	0.08 ± 0.01 ^b^	0.13 ± 0.02 ^a^	0.16 ± 0.02 ^a^
exchangeable Al, g kg^−1^	7.5 ± 0.8 ^a^	6.6 ± 0.8 ^a^	6.6 ± 0.6 ^a^	6.6 ± 0.4 ^a^	7.9 ± 0.8 ^a^

Different letters indicate significant differences at *p* < 0.05.

**Table 3 plants-10-00340-t003:** Mean (± SE) values of physiological traits of cucumber seedlings grown on the soil substrates with lignosulfonate concentration of 0 (0LS), 1 (1LS), 2.5 (2.5LS), 5 (5LS), and 10 (10LS) % under sufficient or low nutrient availability.

Variables	Sufficient Nutrient Availability					Low Nutrient Availability				
	0LS	1LS	2.5LS	5LS	10LS	0LS	1LS	2.5LS	5LS	10LS
Plant DM, g plant^−1^	1.96 ± 0.21 ^a^	2.08 ± 0.12 ^a^	2.03 ± 0.10 ^a^	1.80 ± 0.20 ^ab^	1.53 ± 0.12 ^b^	0.11 ± 0.00 ^c^	0.17 ± 0.02 ^c^	0.18 ± 0.02 ^c^	0.20 ± 0.02 ^c^	0.22 ± 0.02 ^c^
Leaf DM, g plant^−1^	1.11 ± 0.21 ^ab^	1.31 ± 0.07 ^a^	1.28 ± 0.06 ^a^	1.12 ± 0.11 ^a^	0.97 ± 0.06 ^b^	0.03 ± 0.00 ^c^	0.07 ± 0.01 ^c^	0.08 ± 0.01 ^c^	0.09 ± 0.01 ^c^	0.12 ± 0.02 ^c^
Leaf DM:Plant DM	0.64 ± 0.01 ^a^	0.63 ± 0.01 ^a^	0.64 ± 0.01 ^a^	0.63 ± 0.02 ^a^	0.64 ± 0.01 ^a^	0.29 ± 0.02 ^f^	0.40 ± 0.03 ^dc^	0.45 ± 0.04 ^c^	0.48 ± 0.03 ^bc^	0.53 ± 0.03 ^b^
Leaf number plant^−1^	8.25 ± 0.63 ^a^	7.88 ± 0.72 ^a^	7.94 ± 0.32 ^a^	8.14 ± 0.59 ^a^	8.44 ± 0.41 ^a^	2.00 ± 0.17 ^b^	2.17 ± 0.17 ^b^	2.13 ± 0.13 ^b^	1.86 ± 0.26 ^b^	2.22 ± 0.36 ^b^
Total leaf area, cm^−2^	136 ± 18 ^ab^	146 ± 11 ^a^	133 ± 9 ^ab^	133 ± 14 ^ab^	117 ± 12 ^b^	8 ± 1 ^c^	9 ± 1 ^c^	9 ± 1 ^c^	11 ± 1 ^c^	12 ± 1 ^c^
LMA, g m^−2^	36.2 ± 1.2 ^d^	34.0 ± 16 ^d^	42.2 ± 1.6 ^c^	42.3 ± 1.8 ^c^	39.3 ± 1.8 ^d^	44.4 ± 2.6 ^bc^	47.1 ± 1.3 ^bc^	49.2 ± 2.4 ^b^	56.3 ± 2.9 ^a^	59.9 ± 1.4 ^a^
*F*_v_/*F*_m_	0.80 ± 0.00 ^a^	0.80 ± 0.00 ^a^	0.79 ± 0.00 ^a^	0.79 ± 0.00 ^a^	0.80 ± 0.00 ^a^	0.40 ± 0.03 ^b^	0.43 ± 0.03 ^b^	0.43 ± 0.07 ^b^	0.42 ± 0.06 ^b^	0.43 ± 0.07 ^b^
SPAD index	58.4 ± 2.1 ^a^	52.1 ± 1.1 ^b^	49.9 ± 2.1 ^b^	54.8 ± 1.1 ^ab^	53.8 ± 1.8 ^ab^	42.7 ± 2.6 ^c^	37.1 ± 2.1 ^d^	41.9 ± 1.0 ^cd^	43.5 ± 0.7 ^c^	39.5 ± 1.2 ^cd^
*A*_n area_, μmol m^−2^ s^−1^	13.5 ± 0.6 ^a^	11.7 ± 0.8 ^b^	12.4 ± 0.7 ^ab^	12.4 ± 0.45 ^ab^	14.0 ± 0.4 ^a^	0.36 ± 0.05 ^d^	1.55 ± 0.31 ^cd^	1.33 ± 0.19 ^cd^	2.00 ± 0.11 ^c^	1.53 ± 0.27 ^cd^
*A*_n mass_, μmol m^−2^ s^−1^	373 ± 17 ^a^	343 ± 26 ^ab^	295 ± 16 ^b^	293 ± 11 ^b^	376 ± 11 ^a^	9.4 ± 1.5 ^c^	32.9 ± 4.6 ^c^	27.0 ± 3.9 ^c^	35.5 ± 2.0 ^c^	25.5 ± 4.5 ^c^
*g*_s_, mmol m^−2^ s^−1^	133 ± 9 ^ab^	114 ± 15 ^ab^	118 ± 17 ^ab^	114 ± 4 ^b^	140 ± 8 ^a^	32 ± 5 ^c^	41 ± 6 ^c^	39 ± 3 ^c^	51 ± 3 ^c^	45 ± 6 ^c^
Tr, mmol m^−2^ s^−1^	1.93 ± 0.1 ^a^	1.69 ± 0.16 ^a^	1.78 ± 0.20 ^a^	1.69 ± 0.06 ^a^	2.01 ± 0.07 ^a^	0.57 ± 0.10 ^c^	0.69 ± 0.07 ^cb^	0.63 ± 0.04 ^c^	0.92 ± 0.07 ^b^	0.84 ± 0.13 ^bc^
*C*_i_:*C*_a_	0.58 ± 0.01 ^c^	0.55 ± 0.03 ^c^	0.54 ± 0.05 ^c^	0.56 ± 0.01 ^c^	0.59 ± 0.02 ^c^	0.93 ± 0.01 ^a^	0.82 ± 0.02 ^b^	0.84 ± 0.01 ^b^	0.82 ± 0.01 ^b^	0.84 ± 0.01 ^b^
N, g kg^−1^	28.0 ± 3.4 ^b^	40.9 ± 2.2 ^a^	30.3 ± 5.5 ^ab^	28.7 ± 3.9 ^b^	27.4 ± 2.8 ^b^	33.7 ± (5%)	16.0 ± (5%)	14.1 ± (5%)	9.5 ± (5%)	17.0 ± (5%)
P, g kg ^−1^	2.1 ± 0.4 ^b^	3.4 ± 1.0 ^a^	2.6 ± 0.6 ^ab^	2.1 ± 0.3 ^b^	2.9 ± 0.3 ^ab^	1.3 ± (5%)	1.2 ± (5%)	1.7 ± (5%)	0.8 ± (5%)	1.6 ± (5%)
N:P	13.9 ± 1.2 ^a^	11.9 ± 0.7 ^a^	12.3 ± 1.0 ^a^	13.6 ± 0.2 ^a^	9.5 ± 0.6 ^b^	25.7 ± (5%)	25.7 ± (5%)	8.1 ± (5%)	11.2 ± (5%)	10.8 ± (5%)
K, g kg ^−1^	14.6 ± 2.0 ^a^	16.6 ± 1.6 ^a^	17.1 ± 1.7 ^a^	14.3 ± 0.7 ^a^	15.7 ± 2.5 ^a^	10.5 ± (5%)	8.4 ± (5%)	8.7 ± (5%)	8.0 ± (5%)	9.1 ± (5%)
Ca, g kg ^−1^	11.8 ± 0.5 ^b^	14.0 ± 1.1 ^a^	11.5 ± 0.5 ^b^	10.4 ± 0.2 ^bc^	9.5 ± 0.7 ^c^	13.4 ± (5%)	15.8 ± (5%)	12.8 ± (5%)	12.1 ± (5%)	11.6 ± (5%)
Mg, g kg ^−1^	8.4 ± 1.0 ^ab^	9.7 ± 0.5 ^a^	8.2 ± 0.6 ^abc^	6.7 ± 0.4 ^bc^	6.3 ± 0.3 ^c^	5.4 ± (5%)	5.9 ± (5%)	5.2 ± (5%)	4.6 ± (5%)	3.9 ± (5%)
Fe, mg kg^−1^	99.2 ± 16.7 ^a^	92.8 ± 8.4 ^a^	82.6 ± 2.5 ^a^	84.4 ± 5.8 ^a^	90.0 ± 9.8 ^a^	2077 ± (5%)	1300 ± (5%)	378 ± (5%)	493 ± (5%)	304 ± (5%)
Mn, mg kg^−1^	954 ± 37 ^a^	365 ± 137 ^b^	196 ± 13.4 ^bc^	116 ± 27 ^c^	224 ± 10.0 ^bc^	960 ± (5%)	203 ± (5%)	134 ± (5%)	90 ± (5%)	220 ± (5%)
Na, g kg ^−1^	1.02 ± 0.1 ^c^	2.0 ± 0.2 ^c^	3.7 ± 0.3 ^b^	3.8 ± 0.3 ^b^	6.8 ± 0.7 ^a^	3.7 ± (5%)	7.9 ± (5%)	9.0 ± (5%)	10.7 ± (5%)	14.9 ± (5%)
Al, mg kg^−1^	0.11 ± 0.00 ^a^	0.04 ± 0.00 ^ab^	0.02 ± 0.00 ^b^	0.03 ± 0.00 ^b^	0.08 ± 0.00 ^ab^	0.64 ± (5%)	2.31 ± (5%)	1.97 ± (5%)	2.32 ± (5%)	0.10 ± (5%)
SLN, g m^−2^	1.0 ± 0.3 ^bc^	1.5 ± 0.7 ^a^	1.3 ± 0.2 ^ab^	1.2 ± 0.2 ^ab^	1.2 ± 0.1 ^ab^	1.6 ± 0.2 ^a^	0.7 ± 0.0 ^cd^	0.7 ± 0.0 ^df^	0.6 ± 0.0 ^f^	1.0 ± 0.0 ^bc^
SLP, g m^−2^	0.08 ± 0.03 ^bc^	0.13 ± 0.00 ^a^	0.11 ± 0.02 ^ab^	0.09 ± 0.02 ^bc^	0.13 ± 0.01 ^a^	0.06 ± 0.01 ^cd^	0.03 ± 0.00 ^fd^	0.09 ± 0.00 ^bc^	0.05 ± 0.00 ^df^	0.09 ± 0.00 ^bc^
SLK, g m^−2^	533 ± 85 ^cd^	613 ± 61 b^c^	714 ± 67 ^ab^	522 ± 18 ^cd^	712 ± 70 ^a^	499 ± 76 ^cdf^	397 ± 6 ^f^	430 ± 16 ^df^	440 ± 21 ^df^	544 ± 14 ^cd^
SLCa, g m^−2^	472 ± 28 ^cd^	518 ± 35 ^c^	480 ± 18 ^cd^	443 ± 26 ^cd^	413 ± 29 ^d^	637 ± 97 ^b^	745 ± 11 ^a^	632 ± 24 ^b^	664 ± 31 ^b^	690 ± 17 ^ab^
SLMg, g m^−2^	309 ± 45 ^ab^	357 ± 19 ^a^	342 ± 25 ^a^	283 ± 8 ^bc^	281 ± 32 ^bc^	256 ± 39 ^bc^	276 ± 4 ^bc^	255 ± 10 ^c^	254 ± 12 ^c^	231 ± 6 ^c^
SLFe, g m^−2^	3.7 ± 0.8 ^f^	3.4 ± 0.3 ^f^	3.5 ± 0.1 ^f^	3.6 ± 0.3 ^f^	3.9 ± 0.5 ^f^	98.5 ± 15.1 ^c^	61.2 ± 0.9 ^b^	18.6 ± 0.7 ^d^	27.1 ± 1.3 ^c^	18.1 ± 0.4 ^d^
SLMn, g m^−2^	35 ± 1 ^b^	13 ± 59 ^c^	8 ± 1 d^fg^	5 ± 1 ^g^	10 ± 1 ^cdf^	45 ± 7 ^a^	10 ± 0 ^cdf^	7 ± 0 ^fg^	5 ± 0 ^g^	13 ± 0 ^c^
SLNa, g m^−2^	38 ± 6 ^j^	75 ± 6 ^h^	155 ± 12 ^g^	164 ± 21 ^g^	303 ± 52 ^f^	174 ± 27 ^g^	371 ± 6 ^d^	444 ± 17 ^c^	590 ± 28 ^b^	891 ± 22 ^a^

DM, dry mass; LMA, leaf mass per area; *F*_v_/*F*_m_, the maximal quantum yield of PSII photochemistry; SPAD index, leaf chlorophyll concentration; *A*_n area_, area-based net CO_2_ assimilation rate; *A*_n mass_, mass-based net CO_2_ assimilation rate; *g*_s_, stomatal conductance; Tr, transpiration rate; *C*_i_:*C*_a_, the ratio of intercellular to ambient CO_2_ concentration; SLN, specific leaf nitrogen (SLP, SLK, SLCa, SLMg, SLFe, SLMn, SLNa), N (P, K, Ca, Mg, Fe, Mn, Na) concentration in unit leaf area. Different letters indicate significant differences at *p* < 0.05.

**Table 4 plants-10-00340-t004:** Statistical results (*p*) of TWO-WAY ANOVA.

Parameter	Treatment Factor, Interaction		
	Lignosulfonate	Nutrient Availability	Lignosulfonate + Nutrient Availability
Plant DW	0.005 **	<0.001 ***	0.002 **
Leaf DM	0.002 **	<0.001 ***	<0.001 ***
Leaf DM:Plant DM	<0.001 ***	<0.001 ***	0.051 ns
Leaf number	0.316 ns	<0.001 ***	0.324 ns
Total leaf area	0.132 ns	<0.001 ***	0.093 ns
LMA	<0.001 ***	<0.001 ***	<0.001 ***
*F*_v_/*F*_m_	0.999 ns	<0.001 ***	0.689 ns
SPAD index	0.019 *	<0.001 ***	0.257 ns
*A* _n area_	0.084 ns	<0.001 ***	0.002 **
*A* _n mass_	<0.001 ***	<0.001 ***	<0.001 ***
Tr	0.278 ns	<0.001 ***	0.092 ns
*g* _s_	0. 295 ns	<0.001 ***	0.089 ns
*C*_i_:*C*_a_	0.008 **	<0.001 ***	0.182 ns
N	0.569 ns	0.020 *	0.398 ns
P	0.803 ns	0.006 **	0.609ns
K	0.010 *	<0.001 ***	0.010 *
Ca	<0.001 ***	<0.001 ***	<0.001 ***
Mg	<0.001 ***	<0.001 ***	<0.001 ***
Fe	<0.001 ***	<0.001 ***	0.051ns
Mn	<0.001 ***	<0.001 ***	<0.001 ***
Na	<0.001 ***	<0.001 ***	<0.001 ***
Al	<0.001 ***	<0.001 ***	<0.001 ***
SLN	0.012 *	<0.001 ***	<0.001 ***
SLP	<0.001 ***	<0.001 ***	<0.001 ***
SLK	0.051 ns	<0.001 ***	0.210 ns
SLCa	0.017 *	<0.001 ***	0.354 ns
SLMg	<0.001 ***	0.011 *	0.471 ns
SLFe	<0.001 ***	<0.001 ***	<0.001 ***
SLMn	0.072 ns	<0.001 ***	<0.001 ***
SLNa	<0.001 ***	<0.001 ***	<0.001 ***
PNUE	0.146 ns	<0.001 ***	0.017 *
PPUE	0.098 ns	<0.001 ***	0.007 **
PKUE	0.151 ns	<0.001 ***	0.039 *
PCaUE	<0.001 ***	<0.001 ***	<0.001 ***
PMgUE	0.003 **	<0.001 ***	0.006 **
PFeUE	0.675 ns	<0.001 ***	0.614 ns
PMnUE	<0.001 ***	<0.001 ***	<0.001 ***
PNaUE	<0.001 ***	<0.001 ***	<0.001 ***
PWUE	0.019 *	<0.001 ***	0.066 ns
α	0.048 *	<0.001 ***	0.062 ns

Asterisks denote significance levels: * *p* < 0.05, ** *p* < 0.01, *** *p* < 0.001; DM, dry mass; LMA, leaf mass area; *F*_v_/*F*_m_, the maximal quantum yield of PSII photochemistry; SPAD index, leaf chlorophyll content; *A*_n area_ and *A*_n mass,_ area-based and mass-based net CO_2_ assimilation rate; Tr, transpiration rate; *g*_s_, stomatal conductance; *C*_i_:*C*_a_, the ratio of intercellular to ambient CO_2_ concentration; SLN specific leaf nitrogen (SLP, SLK, SLCa, SLMg, SLFe, Slmn, SLNa), N (P, K, Ca, Mg, Fe, Mn, Na) concentration in unit leaf area; PNUE (PPUE, PKUE, PCaUE, PMgUE, PFeUE, PMnUE, PNaUE) photosynthetic N (P, K, Ca, Mg, Fe, Mn, Na) use efficiency; WUE, water use efficiency; α, apparent quantum yield of photosynthesis; ns, not significant.

## Data Availability

Not applicable.

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
