# Peer review of "Photosynthetic Nutrient and Water Use Efficiency of Cucumis sativus under Contrasting Soil Nutrient and Lignosulfonate Levels"

_plants, 2021, doi:10.3390/plants10020340_

Round 1

Reviewer 1 Report

The present study investigated the effects of different lignosulfonate concentrations on the nutrient uptake and photosynthesis of Cucumis sativus under sufficient and low nutrient conditions. They found that lignosulfonate tend to inhibit photosynthetic efficiency of most nutrients under sufficient nutrient conditions but to improve that under low nutrient conditions. This seems interesting. However, I want to say that cucumber grew too poor under the present low nutrient condition on the basis of the very low biomass (Table 2). Thus, the primary limiting factor for plant growth under such poor condition is nutrient deficiency, and this deficiency cannot be completely eliminated by lignosulfonate application. As a result, the beneficial effects of lignosulfonate cannot be observed. Under sufficient nutrient condition, because nutrient supply is enough for cucumber growth, lignosulfonate cannot promote plant growth and nutrient uptake, and even result in some toxic effects. I suggest that the authors add these discussions in the revised manuscript. The below comments are also concerned.

How about the physical and chemical properties of lignosulfonate used in this study? For example, pH and nutritional element concentrations.

Please provide the extraction and analyses methods for the element contents in the soil substrates present in Table 1. Are the data in Table 1 available or total contents of these elements? The nutrient available contents are much more important than total contents for the present study.

How about the effects of LS application on the Fe and Mn contents of soil? They cannot be found in Table 1.

According to the soil pH in Table 1, the used soil is strongly acid (pH 3.89). In such an acid soil, aluminum toxicity may be the main limiting factor for plant growth and photosynthesis. However, the authors did not determine the aluminum concentration of soils and plants.

Author Response

First we would like thank the Reviewers for attentive revision of our manuscript and constructive comments and suggestions.

Let me indicate the modifications made in the manuscript in the light of Reviewer’s comments.

The present study investigated the effects of different lignosulfonate concentrations on the nutrient uptake and photosynthesis of Cucumis sativus under sufficient and low nutrient conditions. They found that lignosulfonate tend to inhibit photosynthetic efficiency of most nutrients under sufficient nutrient conditions but to improve that under low nutrient conditions. This seems interesting. However, I want to say that cucumber grew too poor under the present low nutrient condition on the basis of the very low biomass (Table 2). Thus, the primary limiting factor for plant growth under such poor condition is nutrient deficiency, and this deficiency cannot be completely eliminated by lignosulfonate application. As a result, the beneficial effects of lignosulfonate cannot be observed. Under sufficient nutrient condition, because nutrient supply is enough for cucumber growth, lignosulfonate cannot promote plant growth and nutrient uptake, and even result in some toxic effects.

*     I suggest that the authors add these discussions in the revised manuscript. The below comments are also concerned.

Thank you for this important comment. We added in the Discussion “As outlined in the Introduction we hypothesized that sodium LS application can improve soil nutrient availability and nutrient uptake by plants that, therefore, can positively affect the plant physiological traits. Our hypothesis was based on the early findings that LS can improve soil chemical conditions [9–10] and some physiological traits of plants [14–18]. The agricultural application of LSs, as low-cost and environmental-friendly compounds, could be highly advantageous [6, 10–11] and might promote the effective utilization of wastes of paper and pulp industry. However, we found no evidence of beneficial effects of LS on cucumber seedlings growth, development, photosynthesis, and photosynthetic nutrient and water use efficiency regardless of condition of nutrient availability for plants. Under sufficient nutrient condition, LS did not promote plant nutrient uptake, and even resulted in some toxic effect, as well as, decreased the efficiency of nutrient use at leaf level. Moreover, LS application was not successful to eliminate the negative effects of nutrient deficiency, accompanied by high soil acidity, on plant physiological state, including nutrient and water use efficiency.” Lines 177-190.

*      How about the physical and chemical properties of lignosulfonate used in this study? For example, pH and nutritional element concentrations.

Yes. We have added measured data on acidity and some element content in lignosulfonate used in our study (Table 1).  Moreover, we added “The entire volume of the dry soil was divided into five parts and mixed with sodium lignosulfonate (Kondopoga Pulp and Paper Mill, Karelia, Russia) to achieve its concentration in the soil substrates, equal 0, 1.0, 2.5, 5.0 and 10 % (w/w), designated as 0LS, 1LS, 2.5LS, 5LS and 10LS treatments, respectively. Lignosulfonate used in this study contained 48% of O, 42% of C, 7% of S, 4% of H with ash content of 17% [9].” Lines 262-267.

*      Please provide the extraction and analyses methods for the element contents in the soil substrates present in Table 1. Are the data in Table 1 available or total contents of these elements? The nutrient available contents are much more important than total contents for the present study.

Thank You. We added “Total N concentration in the soil substrates was determined by the Kjeldahl method using Kjeltec analyzer [36], available K and P were extracted with 0.2M HCl and determined using spectrophotometric and flame photometric methods (SF 2000 OKB Spectrum, Saint-Petersburg, Russia) accordingly following Kirsanov’s procedure used in Russia [37]. Soil Ca, Mg, Fe, Mn, Na and Al were extracted with 1MNH4Ac buffered to pH 7 and then determined using spectrophotometric atomic absorption (Shimadzu АА-7000, Kioto, Japan) following the procedure recommended by van Reeuwijk [38].” in the text. Lines 315-322.

Moreover, we indicated total, available or exchangeable contents of soil elements in the Table 2.

*       How about the effects of LS application on the Fe and Mn contents of soil? They cannot be found in Table 1.

We added the data of soil Fe and Mn in the Table 2 and in the section Results. Lines 68-73.

*      According to the soil pH in Table 1, the used soil is strongly acid (pH 3.89). In such an acid soil, aluminum toxicity may be the main limiting factor for plant growth and photosynthesis. However, the authors did not determine the aluminum concentration of soils and plants.

Thank You. We have determined Al content in the soil substrates, as well as, in the leaves. This values are included in the Table 2 and 3, respectively, in the section Results (Lines 68-73) and aluminum toxicity, as the main limiting factor for plant growth and photosynthesis is discussed. Lines 194-203.

The changed text in manuscript is now colored with yellow.

Thanks one more to you!

Best regards,

Reviewer 2 Report

REVIEW for PLANTS-1073126

Photosynthetic nutrient and water use efficiency of Cucumis sativus under contrasting soil nutrient and lignosulfonate level

General impression: The paper presents some data of limited novelty on the use of pulp and paper industry wastes as an alternative source of nutrients. There are several weak points-and if they are not sufficiently responded and elucidated by the authors- I think the paper can not be published in its current form. Another weak part of this ms is the poor English quality. I really encourage the authors to revise their ms, based on the advices of an English native speaker. Finally, the general structure of the ms is very confusing; for example M&M should be placed before the Results and the authors should fully explain (including references) how they determined the different studied parameters. More specifically:    

Abstract

‘Under low nutrient availability, soil lignosulfonate tended to increase nutrient, water and light use efficiency, but it was not successful to eliminate the negative effects of soil nutrient deficiency on plant growth, photosynthetic processes, and efficiency of nutrient use’.

I did not understand if lignosulfonate application under low nutrient availability influenced nutrient use efficiency; it is confusing from this last sentence. Please, rephrase and explain it better.

Introduction

Lines 31-32: ‘…on the concentration of mineral nutrients available in soils’.

Please, rephrase into ‘soil nutrient availability’            

Generally English quality is poor in the Introduction; sometimes it is difficult for the reader to understand the meaning, as for example happens in the second sentence.

Results   

For the title in subparagraph 2.1. I prefer changing the title into: ‘soil pH and nutrient content’ (instead of ‘element content’)

You have to firstly explain the terms ‘SNA’ and ‘LNA’ (where firstly are met in the text) and afterwards you may use them throughout the text.

Paragraph 2.2. is very confusing for the reader, with low linguistic quality. It needs substantial reformation, focusing only on the most important data

Please use ‘TWO-WAY’, instead of ‘two-way’ ANOVA

Why Figure is appeared before the Tables? Since firstly mentioned the Tables in the text, Figure should be put afterwards.

Results should be put after the Materials and Methods. For example it is difficult for the reader to see how you calculated PNUE, PPUE e.t.c.

In the Results section you have to include a small Table (before Table 1) showing the main chemical properties of lignosulfonate

Figure is firstly mentioned in the subparagraph 2.5., thus it should be placed after the Tables.

Materials and Methods

It is not clear in the M&M section how you differentiated ‘sufficient’ and ‘low’ nutrient availability

Subparagraph 4.1.

‘The collected soil was air-dried and sieved with a 2 mm sieve’   For how much time the samples were dried? Under room temperature? Please, include this information.

Subparagraph 4.4.    I think the information contained in this paragraph (plant growth parameters) should be moved after the paragraph 4.2. and before 4.3. (Gas exchange measurements). 

Subparagraph 4.5.       The correct is ‘Chemical analyses’      Please, make the necessary correction.

Generally, M&M is better written, compared to the Results. However, it should be placed after the Introduction and before the Results, since it is impossible for the reader to deeply understand the Results’ section without having all the necessary information on the methodology before.

Discussion

‘At lower levels of supply, Na is beneficial to many species’         Please, justify (with references) this claim; otherwise, please delete this sentence’

Please, fully explain the term ‘LMA’ the first time is appeared in the text.

Although Discussion is not very lengthy, the way it comments the data is quite monotonous and confusing. I think it should be thoroughly reconstructed.

Recommendation to the Editor: The ms discusses some interesting points of pulp and paper industry wastes as an alternative source of nutrients. From this point of view some data merit publication, although the novelty is limited. In addition, there are several weak points throughout the whole text impeding publication of this ms in its current form. Only if the authors would be willing to undertake a major revision of this ms-based on all my remarks-I would recommend the Editor to reconsider it for further processing.    

Author Response

First we would like thank the Reviewers for attentive revision of our manuscript and constructive comments and suggestions.

Let me indicate the modifications made in the manuscript in the light of Reviewer’s comments.

General impression: The paper presents some data of limited novelty on the use of pulp and paper industry wastes as an alternative source of nutrients. There are several weak points-and if they are not sufficiently responded and elucidated by the authors- I think the paper can not be published in its current form. Another weak part of this ms is the poor English quality. I really encourage the authors to revise their ms, based on the advices of an English native speaker. Finally, the general structure of the ms is very confusing; for example M&M should be placed before the Results and the authors should fully explain (including references) how they determined the different studied parameters. More specifically:   

Abstract

‘Under low nutrient availability, soil lignosulfonate tended to increase nutrient, water and light use efficiency, but it was not successful to eliminate the negative effects of soil nutrient deficiency on plant growth, photosynthetic processes, and efficiency of nutrient use’.

*    I did not understand if lignosulfonate application under low nutrient availability influenced nutrient use efficiency; it is confusing from this last sentence. Please, rephrase and explain it better.

Thank you. We rephrased this sentences into ‘The plant biomass, leaf number and area, photosynthesis, and photosynthetic nutrient, water, and light use efficiency were significantly reduced with a nutrient deficiency. Sodium lignosulfonate application was not successful to eliminate the negative effects of nutrient deficit on cucumber plant.‘ Lines 20-24.

Introduction

*    Lines 31-32: ‘…on the concentration of mineral nutrients available in soils’. Please, rephrase into ‘soil nutrient availability’ 

  Thank you. Done.

Generally English quality is poor in the Introduction; sometimes it is difficult for the reader to understand the meaning, as for example happens in the second sentence.

English is corrected throughout the text. The second sentence is changed to “The wide range of soils are noted for low nutrient-supplying capacity, and for agricultural purposes they need to be improved with commercial high-cost fertilizers.’ Instead of ‘In their native state a wide range of soils are low in nutrient capacity, thus, for agricultural purposes these soils, as a rule, need commercial high-cost fertilizer applications.’ Lines 29-31.

Results  

*    For the title in subparagraph 2.1. I prefer changing the title into: ‘soil pH and nutrient content’ (instead of ‘element content’)

Thank you. Done.

*    You have to firstly explain the terms ‘SNA’ and ‘LNA’ (where firstly are met in the text) and afterwards you may use them throughout the text.

Yes, you are right. We changed “With the decline in nutrient availability from SNA to LNA,…” to “With the decline in nutrient availability from sufficient nutrient availability (SNA) to low nutrient availability (LNA),…” Lines

*    Paragraph 2.2. is very confusing for the reader, with low linguistic quality. It needs substantial reformation, focusing only on the most important data

We changed this paragraph to ‘The dry biomass, leaf number and area were significantly lower in seedlings grown under condition of low nutrient availability (LNA) than sufficient nutrient availability (SNA) (Table 3). By contrast, the leaf mass per unite leaf area (LMA) were significantly higher in LNA grown seedlings compared to their SNA grown counterparts. The LS application caused a significant decrease in total plant and leaf biomass only in 10LS seedlings grown under SNA. For LNA grown seedlings, LS had no significant effect on biomass accumulation and its allocation into organs as well as leaf number and area. The LMA values increased following the increase in substrate LS content regardless of nutrient conditions.’ Lines 79-86.

*    Please use ‘TWO-WAY’, instead of ‘two-way’ ANOVA

Thank you. Done.

*    Why Figure is appeared before the Tables? Since firstly mentioned the Tables in the text, Figure should be put afterwards.

The Figure is moved. Thank you.

*    Results should be put after the Materials and Methods. For example it is difficult for the reader to see how you calculated PNUE, PPUE e.t.c.

According to the Instructions for Authors of Plants the section Materials and Methods should be put after the section Discussion. https://www.mdpi.com/journal/plants/instructions

*    In the Results section you have to include a small Table (before Table 1) showing the main chemical properties of lignosulfonate

Thank you. Done. Table 1.

*    Figure is firstly mentioned in the subparagraph 2.5., thus it should be placed after the Tables.

Yes. Done.

Materials and Methods

*    It is not clear in the M&M section how you differentiated ‘sufficient’ and ‘low’ nutrient availability

Thank you. We replaced ‘For the first bloc of the pots, the nutrient solution (based on 1 g l-1 Ca(NO3)2, 0.25 g l-1 KH2PO4, 0.25 g l-1 MgSO4 7H2O, 0.25 g l-1 KNO3, a trace quantity of FeSO4 and pH 6.2−6.4, EC 2.0 mS cm-1), and for the second one, water was supplied every two days. The first bloc was designated as grown under the condition of sufficient nutrient availability (SNA treatment) and the second one was grown under low nutrient availability (LNA treatment).’

 with ‘The pots of each LS treatment were divided into two blocs. For the first bloc of the pots, the nutrient solution composed of 1 g l-1 Ca(NO3)2, 0.25 g l-1 KH2PO4, 0.25 g l-1 MgSO4 7H2O, 0.25 g l-1 KNO3, a trace amount of FeSO4 with pH 6.2−6.4 and EC 2.0 mS cm-1 was supplied every two days. For the second bloc of the pots, water was supplied also every two days. The first bloc of plants was designated as plants grown under the condition of sufficient nutrient availability (SNA) and the second one was grown under low nutrient availability (LNA). Lines 279-284.

Subparagraph 4.1.

*    ‘The collected soil was air-dried and sieved with a 2 mm sieve’   For how much time the samples were dried? Under room temperature? Please, include this information.

We added this information. “The collected soil was air-dried at 21-23°C for 14 days and sieved with a 2 mm sieve.” Line 262.

*    Subparagraph 4.4.    I think the information contained in this paragraph (plant growth parameters) should be moved after the paragraph 4.2. and before 4.3. (Gas exchange measurements).

Done.

*    Subparagraph 4.5.       The correct is ‘Chemical analyses’      Please, make the necessary correction.

Thank you. Done.

Generally, M&M is better written, compared to the Results. However, it should be placed after the Introduction and before the Results, since it is impossible for the reader to deeply understand the Results’ section without having all the necessary information on the methodology before.

Discussion

*    ‘At lower levels of supply, Na is beneficial to many species’         Please, justify (with references) this claim; otherwise, please delete this sentence’

We deleted this part of the sentence. Line 210.

*    Please, fully explain the term ‘LMA’ the first time is appeared in the text.

Thank you. Done. Line 81.

*    Although Discussion is not very lengthy, the way it comments the data is quite monotonous and confusing. I think it should be thoroughly reconstructed.

Yes, you are right. So, the section Discussion is thoroughly reconstructed now.  

The changed text in manuscript is now colored with yellow.

Thanks one more to you!

Best regards,

Reviewer 3 Report

The manuscript of Ikkonen and colleagues reports some applications of lignosulfonate as a natural fertilizer of Cucumis sativus. They evaluate some physiological parameters of the plant and the soil after treatment with different dosages of lignosulfonate.

The manuscript is clear and the initial hypotheses are proven by the results presented and discussed by the authors. Furthermore a correct statistical analysis was applied.

However, there are some observations that the authors should comment on.

1) One of the measurable effects due to a change in the constituents of the medium are secondary metabolites. The authors should discuss the effects on the production of secondary metabolites of Cucumis sativus;

2) It is necessary that the authors discuss the large-scale ecological effects of the use of lignosulfonate, also highlighting problems associated with its use;

3) The authors should indicate the origin of the sodium lignosulfonate used for the experiments. The chemical nature of the compound used is not clear.

4) Finally some manuscripts should be cited in the references:

Quantitative and Qualitative Evaluation of Sorghum bicolor L. under Intercropping with Legumes and Different Weed Control Methods, https://doi.org/10.3390/horticulturae6040078

Germination and Seedling Growth Responses of Zygophyllum fabago, Salsola kali L. and Atriplex canescens to PEG-Induced Drought Stress, 10.3390/environments7120107

Author Response

First we would like thank the Reviewers for attentive revision of our manuscript and constructive comments and suggestions.

Let me indicate the modifications made in the manuscript in the light of Reviewer’s comments.

1) One of the measurable effects due to a change in the constituents of the medium are secondary metabolites. The authors should discuss the effects on the production of secondary metabolites of Cucumis sativus;

Thank You. You are right, that changed growth condition affectes a wide range of physiological processes including the production of secondary metabolites. Yes, they are measurable parameters, but in this study we did not determine the secondary metabolites because the objective was to evaluate the responses of photosynthetic processes, exactly photosynthetic nutrient use efficiency. We know that the production of secondary metabolites is very important for plant metabolism, so we are sure that the investigation of responses of these processes on both LS and soil nutrient condition will be an objective of our near future.

2) It is necessary that the authors discuss the large-scale ecological effects of the use of lignosulfonate, also highlighting problems associated with its use;

Thank You. The section Discussion is thoroughly reconstructed now and this problem is discussed.  For example, we added in the text “The LS application dramatically increased soil and leaf Na content (Table 2, 3). Adding high concentrations of LS to soils is likely to cause sodium salinization of these soils.  Appearing at high concentrations, Na ions can inhibit the main physiological processes in plants due to osmotic stress or ionic effects [26]. The increase in cellular Na can affect the acquisition and distribution of essential nutrients such as N, K, and Ca [27], and, also Mg, as our results have shown (Table 3). For the seedlings grown under sufficient nutrient condition, the LS-related decrease of photosynthetic efficiency of using of most studied nutrients is likely to be associated with the strong increase of leaf Na content.” Lines 208-219.

3) The authors should indicate the origin of the sodium lignosulfonate used for the experiments. The chemical nature of the compound used is not clear.

Yes. Done. We added Table 1. The pH value and element content in the sodium lignosulfonate used in the study. Lines 75-77.

Also we added ‘The entire volume of the dry soil was divided into five parts and mixed with sodium lignosulfonate (Kondopoga Pulp and Paper Mill, Karelia, Russia) to achieve its concentration in the soil substrates, equal 0, 1.0, 2.5, 5.0 and 10 % (w/w), designated as 0LS, 1LS, 2.5LS, 5LS and 10LS treatments, respectively. Lignosulfonate used in this study contained 48% of O, 42% of C, 7% of S, 4% of H with ash content of 17% [9].’ Lines 262-267.

4) Finally some manuscripts should be cited in the references:

Done.

The changed text in manuscript is now colored with yellow.

Thanks one more to you!

Best regards,

Round 2

Reviewer 1 Report

The manuscript has been revised according to the comments.

Author Response

Thank You!

Best regards,

Reviewer 2 Report

Despite the efforts made by the authors to ameliorate the quality of their ms, some problems (such as the low English quality and the monotonous, unecessary, repetition of the results in the Discussion) still exist (although to lesser extent, compared to the previous version of the ms). A good Discussion should be focused only on the comparison only of the most important results to those of other researchers, as well as to their successful commentation. There is no need to repeat all the data without reason (this was done in the Results section). 

Thus, a new effort is needed, in order to reconsider the ms for publication.  

Author Response

Thank You for attentive revision of our manuscript.

Let me indicate the modifications made in the last version of the text.

Despite the efforts made by the authors to ameliorate the quality of their ms, some problems (such as the low English quality and the monotonous, unecessary, repetition of the results in the Discussion) still exist (although to lesser extent, compared to the previous version of the ms). A good Discussion should be focused only on the comparison only of the most important results to those of other researchers, as well as to their successful commentation. There is no need to repeat all the data without reason (this was done in the Results section). 

* Thus, a new effort is needed, in order to reconsider the ms for publication.

Thank You. We rewrote most of the Discussion. We removed the repetitions of the results and focused on the comparison of our results with those of other authors. New comments added to the Discussion.

The changed part of the Discussion is colored with yellow.

Thanks one more to you!

Best regards,
